# Using prior information from humans to prioritize genes and gene-associated variants for complex traits in livestock

Biaty Raymond[1,2]*, Loic Yengo[3], Roy Costilla[4], Chris Schrooten[5], Aniek C. Bouwman[1], Ben J. Hayes[4], Roel F. Veerkamp[1‡], Peter M. Visscher[3‡]

**1** Animal Breeding and Genomics, Wageningen University and Research, Wageningen, The Netherlands, **2** Biometris, Wageningen University and Research, Wageningen, The Netherlands, **3** Institute for Molecular Bioscience, University of Queensland, St. Lucia, Australia, **4** Queensland Alliance for Agriculture and Food Innovation, University of Queensland, St. Lucia, Australia, **5** CRV BV, Arnhem, The Netherlands

‡ These authors jointly supervised this work.
* biaty.raymond@wur.nl

**Data Availability Statement:** The (GWAS) summary statistics data for human height and cattle stature have been published previously. Genotype and phenotype data for a small cattle

## Abstract

Genome-Wide Association Studies (GWAS) in large human cohorts have identified thousands of loci associated with complex traits and diseases. For identifying the genes and gene-associated variants that underlie complex traits in livestock, especially where sample sizes are limiting, it may help to integrate the results of GWAS for equivalent traits in humans as prior information. In this study, we sought to investigate the usefulness of results from a GWAS on human height as prior information for identifying the genes and gene-associated variants that affect stature in cattle, using GWAS summary data on samples sizes of 700,000 and 58,265 for humans and cattle, respectively. Using Fisher's exact test, we observed a significant proportion of cattle stature-associated genes (30/77) that are also associated with human height (odds ratio = 5.1, $p$ = 3.1e-10). Result of randomized sampling tests showed that cattle orthologs of human height-associated genes, hereafter referred to as candidate genes (C-genes), were more enriched for cattle stature GWAS signals than random samples of genes in the cattle genome (p = 0.01). Randomly sampled SNPs within the C-genes also tend to explain more genetic variance for cattle stature (up to 13.2%) than randomly sampled SNPs within random cattle genes (p = 0.09). The most significant SNPs from a cattle GWAS for stature within the C-genes did not explain more genetic variance for cattle stature than the most significant SNPs within random cattle genes ($p$ = 0.87). Altogether, our findings support previous studies that suggest a similarity in the genetic regulation of height across mammalian species. However, with the availability of a powerful GWAS for stature that combined data from 8 cattle breeds, prior information from human-height GWAS does not seem to provide any additional benefit with respect to the identification of genes and gene-associated variants that affect stature in cattle.

population, which we used to validate some of the results in this study were obtained from CRV BV, a commercial cattle breeding company. As such, the company cannot make these data publicly available.

**Funding:** The author(s) received no specific funding for this work.

**Competing interests:** The authors have declared that no competing interests exist.

## Author summary

Provided that there is a similarity in the biological control of complex traits in different species, the results of genome-wide association study for a given trait in a model species can be utilized as prior information to identify genes and gene-associated variants that affect the same trait in a non-model species. Here we report that the number of genes found to be affecting height in both humans and cattle is more than can be expected by chance, suggesting a similarity in the biology of height in both species. In addition, genes that are associated with height in humans were found to be highly enriched with variants that are strongly associated with height in cattle. Using height as a model trait, we demonstrated that across-species prior information cannot compete with a within-species prior information, with respect to the identification of genes and the gene-associated variants that affect a complex trait. Across-species prior information can be useful, however, when within-species prior information is either unavailable or poor.

## Introduction

Genome-wide association studies (GWAS) have become popular and powerful tools for identifying genes involved in complex traits aetiology, and for investigating the genetic architecture of such traits in animal and plant species. In humans for example, GWAS have been used to identify thousands of risk loci for complex diseases such as heart diseases and psychiatric disorders, and have led to the discovery of potential drug targets [1, 2]. In plant and livestock species, the focus of GWAS has primarily been, to identify variants or genomic regions that underlie complex and economically important traits [3, 4]. Ultimately, the ambition, with respect to GWAS, is the identification of a subset of genomic variants that together explain the total genetic variance for complex traits.

The ability of GWAS to identify variants that are associated with complex traits depends on sample size. Large sample size is especially important for identifying rare variants with medium-sized effects and common variants with small effects. A classic example of the importance of sample size in GWAS is the trajectory of the number of loci that have been associated with human height over the past decade. That number rose from ~40 with a sample size between 13,000 to ~ 30,000 in 2008 [5–7], to 180 with a sample size of 183,727 in 2010 [8]. By 2014, the number of loci associated with human height had increased to 697 with a sample size of 253,288 [9], while the latest GWAS for human height from 2018 reported 3,290 near-independently associated loci, with a sample size of ~ 700,000 [10]. It should be noted that the increase noticed in the past five years also reflects improvements in data analysis methods [11]. It is predicted that as sample sizes continue to increase, the number of loci reliably associated with human height will continue to increase [2, 10].

While GWAS sample sizes have continued to increase in humans, the same is not true for livestock species. While there are millions of animals from livestock species that are regularly genotyped around the world, the data are rarely shared or combined for the purpose of GWAS, given that the data are commercially sensitive and proprietary. Up until recently, for example [12–15], most GWAS in livestock have been carried out with cohort level data, with samples sizes varying from a few hundreds to a few thousands [4]. While GWAS in livestock have resulted in a number of trait-associated loci, mainly those with large effect sizes, the joint effect of the discovered loci explain only small proportions of the total genetic variance for the traits of interest [16]. Given the sample-size limitation in most GWAS in livestock, it would be of scientific interest to know if the results of GWAS studies for complex traits in humans can

be used as prior information to identify the genes and gene-associated variants that affect equivalent traits in livestock species.

The idea to utilize GWAS results for a given trait across species presupposes that the genes underlying the trait are conserved across species. There have been reports of genes and mutations affecting height in humans that are found to affect height in other mammalian species, see [17] for a review. In their study, Pryce *et al.* [18] found that variants within 55 cattle genes, whose orthologs are associated with height in humans, were significantly associated with stature in cattle. The result of Pryce *et al.* [18] is further evidence to suggest that the same genes may be underlying height in humans and cattle. Thus, instead of using a within-species GWAS to identify genes and gene-associated variants for complex traits as was done in [12], a GWAS results for the same trait but from a different species could be utilized. With the availability of large sample sizes for GWAS and whole genome sequence data in both humans and cattle, we hypothesised that the evidence for similarity in the biology of height in humans and cattle will become stronger, and that use of GWAS results for height in humans as prior information to identify stature genes in cattle will become feasible. The objective of this study was to test this hypothesis. We therefore investigate the usefulness of GWAS results on human height as a prior information for identifying genes and gene-associated variants that affect stature in cattle. For this, we used GWAS summary data on samples sizes of ~700,000 and 58,265 for humans and cattle, respectively. We find that prior information from human height GWAS is beneficial for identifying genes and gene-associated variants affecting stature in cattle, only when cattle prior information is not available.

## Results

### Orthologous genes in humans and cattle are highly correlated in size but not in LD score or gene-effect on height

From the UMD3.1 cattle genome build, we identified 13,742 genes that have orthologs in humans and passed our quality control (QC). Within each species, we estimated the correlation between gene size and gene LD score, defined as the mean LD score of variants in the vicinity of the gene (Methods). We observed no significant correlation between gene size and gene LD scores across the 13,742 orthologous genes selected within the human genome (r = 0.0029, $p$ = 0.74), and a significant correlation between gene size and gene LD scores within the cattle genome (r = 0.028, $p$ = 0.001). We observed a significant positive correlation in gene-size (r = 0.85, $p$ = 2.2x10$^{-16}$) but no significant correlation in gene LD scores (r = 0.012, $p$ = 0.16) between human genes and their cattle orthologs. Furthermore, we estimated the correlation between the effect of orthologous genes on height in humans and cattle, as a high correlation may be an indication that the genes perform similar function across species. To do this, we estimated the aggregated effect of all the SNPs within each orthologous gene on the phenotype (human height and cattle stature), which we refer to as gene-effect. The estimate of correlation between the effect of the orthologous genes on height was low, but significant (r = 0.12, $p$ = 2.2x10$^{-16}$).

### Height-associated genes in humans and cattle significantly overlap

To address if the same genes underlie height in humans and cattle, we estimated the proportion of orthologous genes in or close to GWAS peaks for human height that are also located in or close to GWAS peaks for stature in cattle, and performed Fisher's exact test to check if the proportion of overlap is more than can be expected by chance. We identified 77 cattle stature-associated orthologous genes based on the results of [12]. Out of the 77 cattle orthologous genes, 10 were also found to be associated with human height based on the result of [9], Fig

1A. The proportion 10/77 is more than can be observed by chance based on the results of Fisher's exact test (odds ratio = 5.5, p = $3.7 \times 10^{-5}$). The contingency table for the Fisher's exact test is provided in the S1 Table. Details about the 10 overlapping genes are provided in S2 Table The overlap was stronger (30/77) between the 77 cattle genes and human height-associated genes identified based on the results of [10], Fig 1B. The human height-associated genes were identified as those that were overlapping with or nearby the near-independent significant SNPs from the meta-GWAS. The proportion 30/77 is more than expected by chance based on the result of Fisher's exact test (odds ratio = 5.1, p-value = $3.1 \times 10^{-10}$). The contingency table for the Fisher's exact test is provided in the S3 Table, and details about the 30 overlapping genes are provided in S4 Table. To check if the average marker density in the 10kbp windows of association with lead SNPs is higher than the average marker density of genes in both species, we plotted the distribution of average marker density (total number of SNPs divided by length in kbp) of the 10kbp windows against the distribution of average marker density of genes in both species (S1 Fig). In general, we observed very similar distributions of average marker density between the windows of interest and genes. Furthermore, the result fisher's exact test (S5 Table) showed no evidence of non-random association in marker density of genes in cattle and humans (p = 0.87), indicating that the estimates of overlap of height genes in cattle and humans is less likely to be influenced by marker density.

It is worth noting that the 77 orthologous cattle stature-associated genes identified from [12], are among the largest in size in both the cattle and human genomes (Fig 2), which could inflate the likelihood of overlap. Therefore, we implemented a gene size-stratified Cochran-Mantel-Haenszel test to correct our estimates of overlap for potential bias. Our result show that accounting for gene-size still results in a significant overlap (p-value = $2.4 \times 10^{-9}$). The contingency table for the Cochran-Mantel-Haenszel test is provided in S6 Table.

## Cattle stature-associated variants are enriched in human height-associated genes

To further check if human height-associated genes also affect stature in cattle, we tested if cattle genes prioritized based on information from a GWAS on human height are more enriched

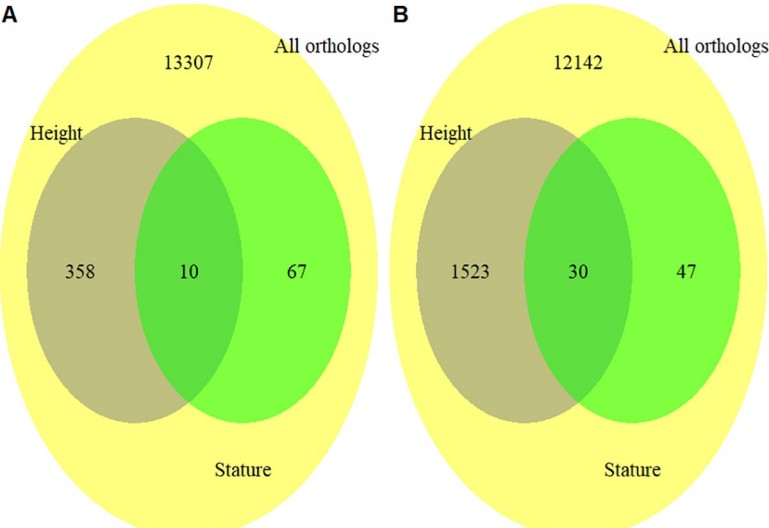

**Fig 1.** Overlap between orthologous genes associated with height in humans based on Wood *et al.* [9] that are also associated with stature in cattle based on Bouwman *et al.* [12] (Panel A), and overlap between orthologous genes associated with height in humans based on Yengo *et al.* [10] that are also associated with stature in cattle based on Bouwman *et al.* [12] (Panel B).

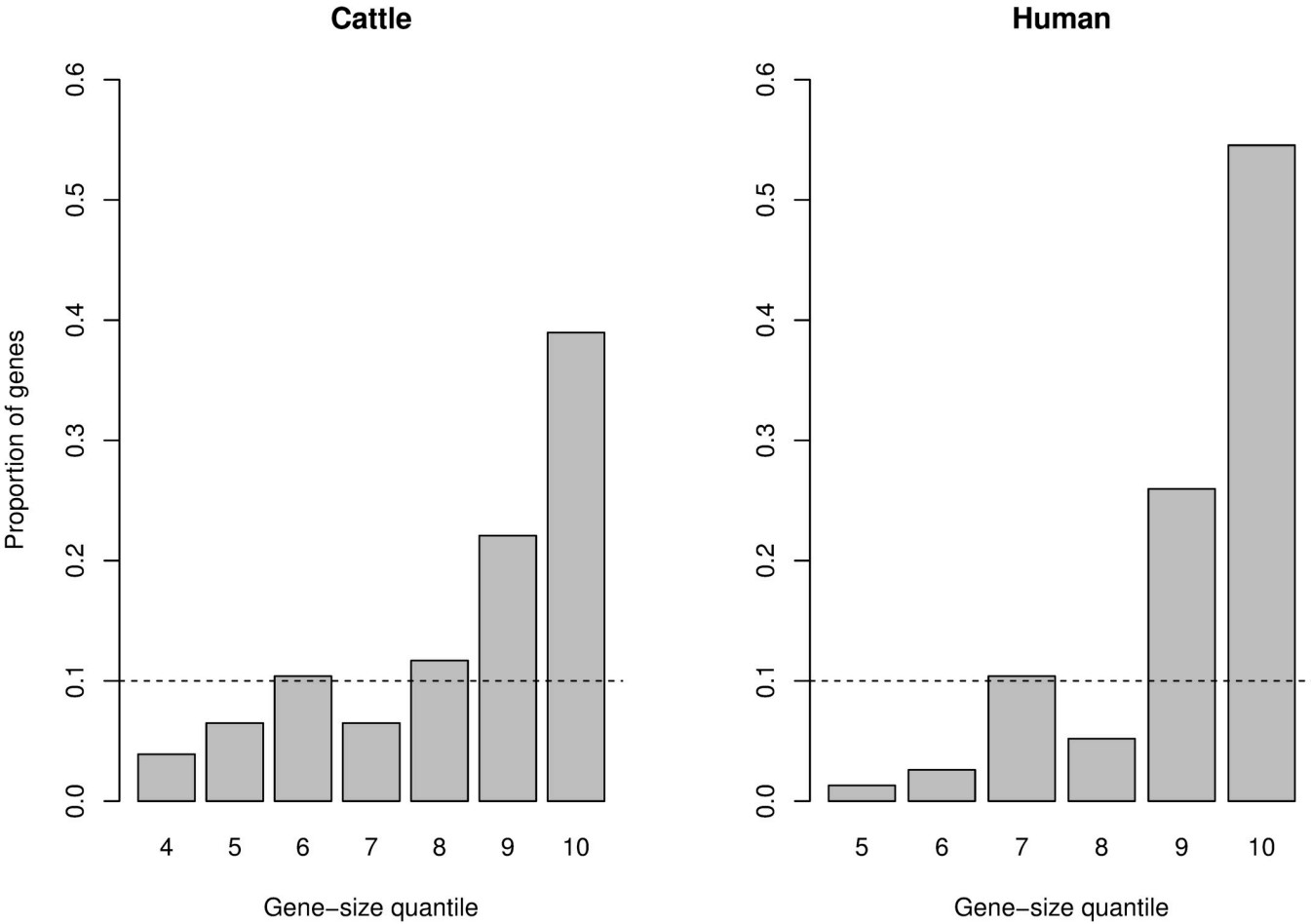

**Fig 2. Distribution of gene-size (in cattle and humans) for the 77 orthologous genes that overlap with or are within 10kbp of the lead SNPs from Bouwman *et al*. [12].** Genes were stratified into 10 quantiles based on gene size, with the shortest genes in 1 and the longest genes in 10.

for GWAS signals for cattle stature than random sets of genes from the cattle genome. For this purpose, we utilized two sets of human height-associated genes. The first set comprised of 370 orthologous genes that were identified by [10] using the summary-based Mendelian Randomization (SMR) analysis [19]. The SMR analysis combines gene-expression data from multiple tissues and summary level GWAS data to identify genes whose expression level is associated with the trait. We identified the cattle orthologs of the SMR-based human genes that we refer to as SMR candidate (SMR-C) genes. The result of randomized sampling test (Methods) showed that the number of near-independently significant SNPs for cattle stature within and around the 370 SMR-C genes are not more than can be expected within and around random sets of cattle genes matched on size, SNP density and mean LD score (Fig 3). The second set of human height-associated genes we used comprised of 1,184 orthologous genes that overlap with the 3,290 near-independently significant SNPs affecting human height identified through a conditional and joint effect (COJO) analysis in [10]. The cattle orthologs of the COJO-based human genes were identified and referred to as COJO candidate (COJO-C) genes. We found that the number of near-independent SNPs for stature in cattle within and around the 1,184 COJO-C genes is more than can be observed within and around random sets of cattle genes matched on size, SNP density and mean LD score (Fig 4).

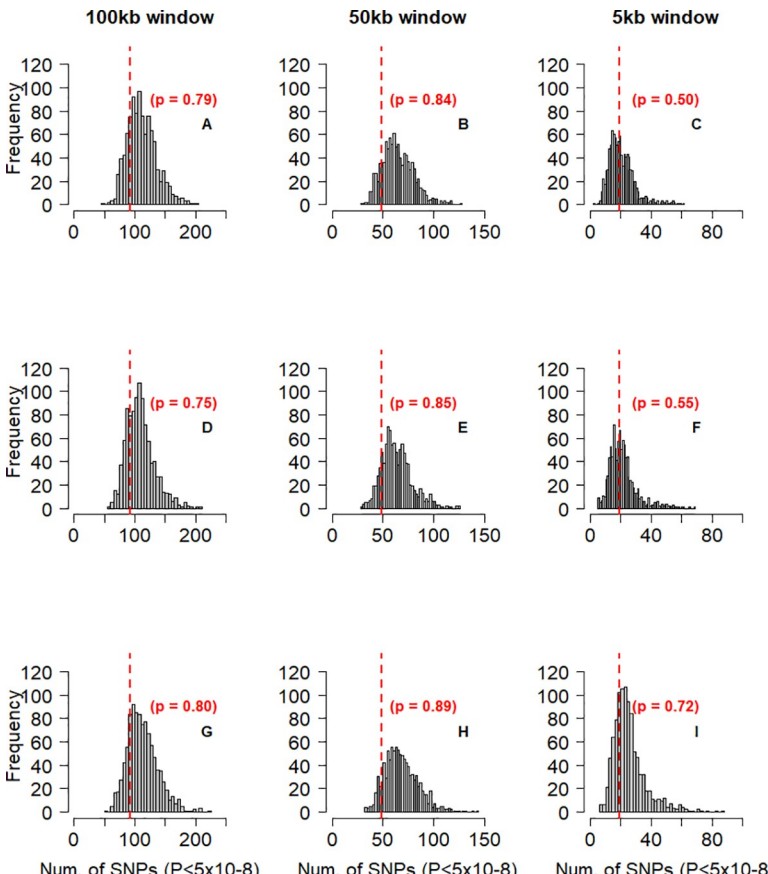

**Fig 3. Frequency distribution of the number of SNPs with near-independent significant effect on stature (P < 5x10-8) in 1,000 control tests for SMR-C genes.** In each test, 370 cattle genes were sampled from a total of 13,865 genes that have human orthologs, and near-independent significant SNPs within and 100kbp (Column 1 – panels A,D,G), 50kbp (Column 2 – panels B,E,H) or 5kbp (Column 3 – panels C,F,I) either sides of the sampled genes were identified. The genes were sampled using stratified random sampling, with the genes stratified based on gene-size (Row 1 – panels A,B,C), number of SNPs in a gene (Row 2 – panels D,E,F) or gene LD score (Row 3 – panels G,H,I). The vertical red lines represents the number of near-independent significant SNPs (P < 5x10-8) associated with stature, identified within and around the 370 cattle (SMR-C) candidate genes that have been shown to be associated with height in humans. The p value represents the proportion of samples (out of 1,000) that have equal or higher level of enrichment for GWAS signals than the SMR-C genes.

The difference in sample size (370 vs 1,184) between the SMR-C and the COJO-C genes may result in the observed difference in the significance of enrichment for cattle stature GWAS signals. However, we found that with the same sample size of 370, the COJO-C genes are more enriched for GWAS signals than the SMR-C genes or any random set of 370 genes matched based on gene-size (Fig 5). This indicates that the results observed for the SMR-C genes is not just an issue of power (small gene sample size).

The SMR analysis uses gene-expression data from multiple tissues. Given that not all tissues are relevant for a given trait, it is critical that gene-expression data from a trait-relevant tissue is used. Hence, we tested if the level of enrichment for GWAS signals in the SMR-C genes will be more than can be expected from random sets of genes if the selection criteria were to be more stringent. In particular, we required that the human ortholog of the gene must show evidence of differential gene expression in three or more tissues. With this selection criterion, we identified a subset of 121 (SMR-MT-C) genes instead of 370. Our result show that the level of enrichment for GWAS hits nearby the 121 SMR-MT-C genes is more than can be expected

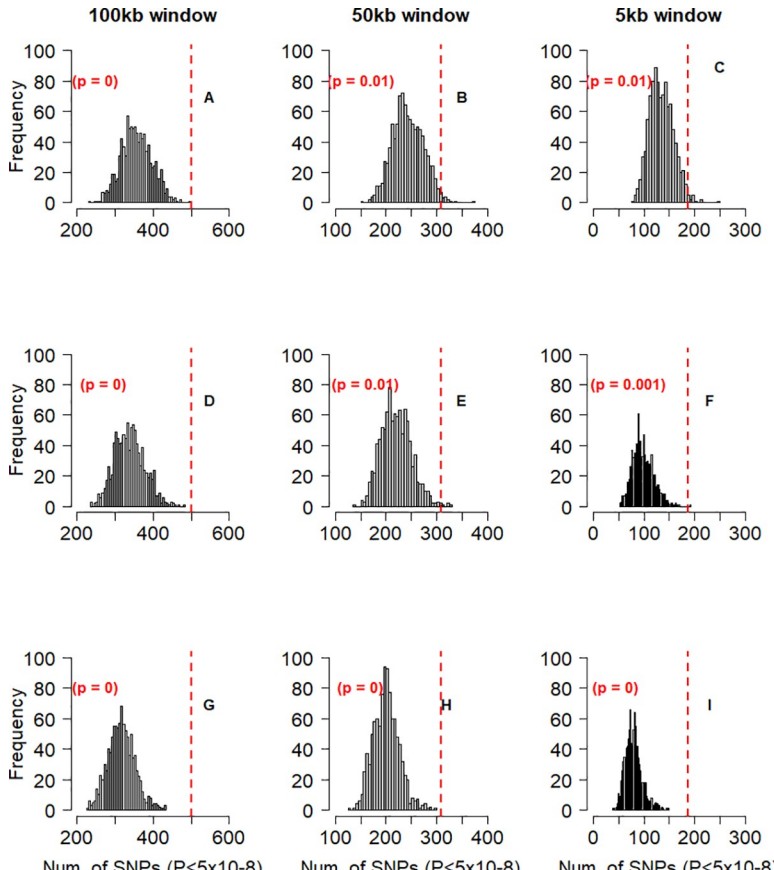

**Fig 4. Frequency distribution of the number of SNPs with near-independent significant effect on stature (P < 5x10-8) in 1,000 control tests for COJO-C genes.** In each test, 1,184 cattle genes were sampled from a total of 13,865 genes that have human orthologs, and near-independent significant SNPs within and 100kbp (Column 1 – panels A,D,G), 50kbp (Column 2 –panels B,E,H) or 5kbp (Column 3 –panels C,F,I) either sides of the sampled genes were identified. the genes were sampled using stratified random sampling, with the genes stratified based on gene-size (Row 1 –panels A,B,C), number of SNPs in a gene (Row 2 –panels D,E,F) or gene LD score (Row 3 –panels G,H,I). The vertical red lines represents the number of near-independent significant SNPs (P < 5x10-8) associated with stature, identified within and around the 1,184 cattle (COJO-C) candidate genes that have been shown to be associated with height in humans. The p value represent the proportion of samples (out of 1,000) that have equal or higher level of enrichment for GWAS signals than the COJO-C genes.

from any set of 121 randomly sampled genes, with p = 0.02 (Fig 6). The p values in the randomized sampling test were calculated as the proportion of randomly sampled gene-sets that had equal or higher level of enrichment for GWAS hits as compared to the candidate gene sets.

## Accuracy of cattle variants selection using within- and across-species prior information

Next, we quantified the proportion of stature variance explained by pre-selected cattle SNPs based on GWAS results for height in humans and stature in cattle. First, we tested the hypothesis that cattle SNPs that are pre-selected based on two information sources 1) genes identified from GWAS results for human height, and 2) Cattle stature GWAS significance level, will explain more genetic variance for stature than SNPs that are pre-selected based on only cattle stature GWAS significance level. To test this hypothesis, we identified the near-independent

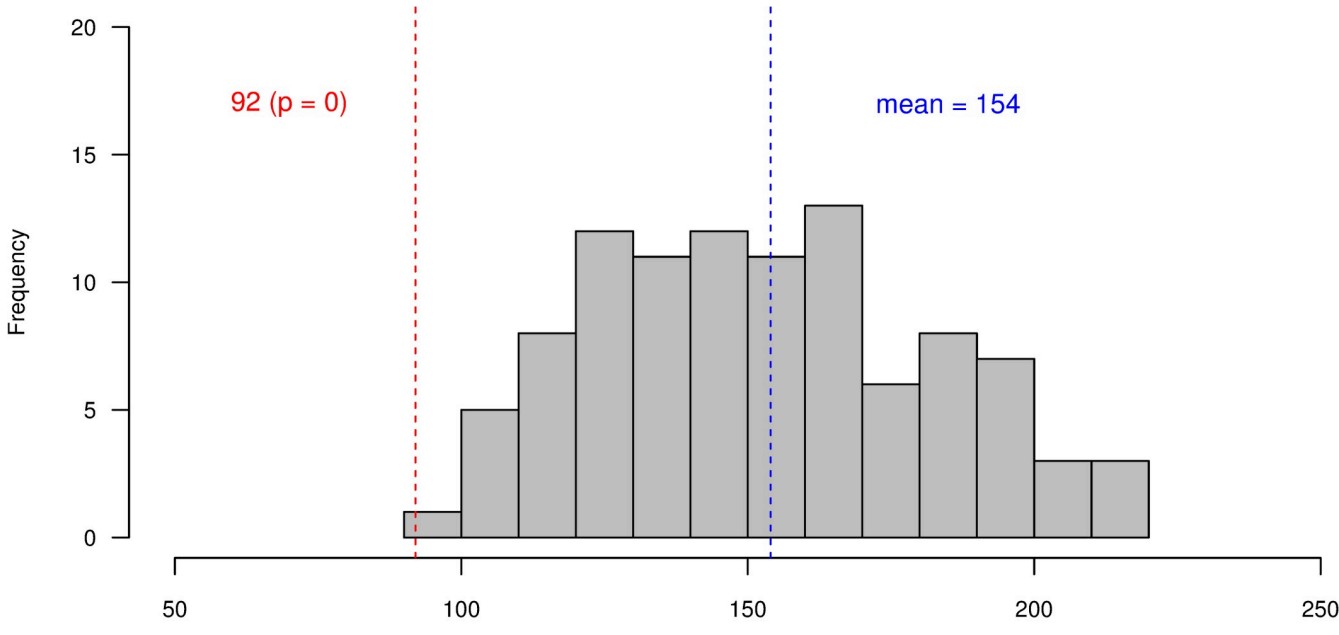

**Fig 5. Frequency distribution of the number of SNPs with near-independent significant effect on stature (P < 5x10-8) in 100 control tests.** In each test, 370 out of 1,184 cattle (COJO-C) genes were sampled. The number of near-independent significant SNPs within and 100kbp either side of the sampled genes were identified and compared to the number obtained within and 100kbp either side of the 370 SMR-C genes. The vertical red dashed line represents the number of near-independent significant SNPs (P < 5x10-8) associated with stature, identified within and 100kbp either side of the 370 SMR-C genes, that have been shown to be associated with height in humans through an SMR analysis. The p value represent the proportion of samples (out of 100) that have equal or lower level of enrichment for GWAS hits than the SMR-C genes.

significant SNPs within 100kbp either side of the SMR-C and COJO-C genes, that also segregate in a New Zealand Holstein (NZH) cattle population: (80 SMR-C SNPs, 391 COJO-C SNPs). The NZH population was not included in the cattle meta-GWAS analysis [12] and is therefore ideal as a validation population. It must be noted, however, that some of the bulls in the NZH population are descendants of bulls from the Dutch Holstein population which was included in the meta-GWAS. The result of randomized sampling test showed that the identified candidate SNPs do not explain more genetic variance for stature in NZH population than the most significant SNPs for cattle stature within randomly sampled cattle genes (Fig 7).

Secondly, we tested the hypothesis that cattle SNPs pre-selected using GWAS result for human height will explain more genetic variance for stature, than random set of SNPs from random cattle genes. To test the hypothesis, we randomly sampled 5 SNPs per gene to avoid having a large number of SNP in a set. For the SMR-C genes, this resulted in 1,777 unique SNPs that were segregating in NZH. For the COJO-C genes, the number of unique SNPs extracted was 5,781. Although the results are not statistically significant, we found that in ~ 90% of time, the candidate SNPs explain more genetic variance for stature in NZH population than randomly sampled SNPs from random sets of genes from the cattle genome (Fig 8).

## Discussion

A general expectation with respect to conserved orthologous genes is that they perform the same or very similar functions in different species and thus the functions of the genes in one species could be inferred to another, especially when such functions are critical for evolutionary fitness [20–22]. Several experiments in the past have reported a similarity in the function

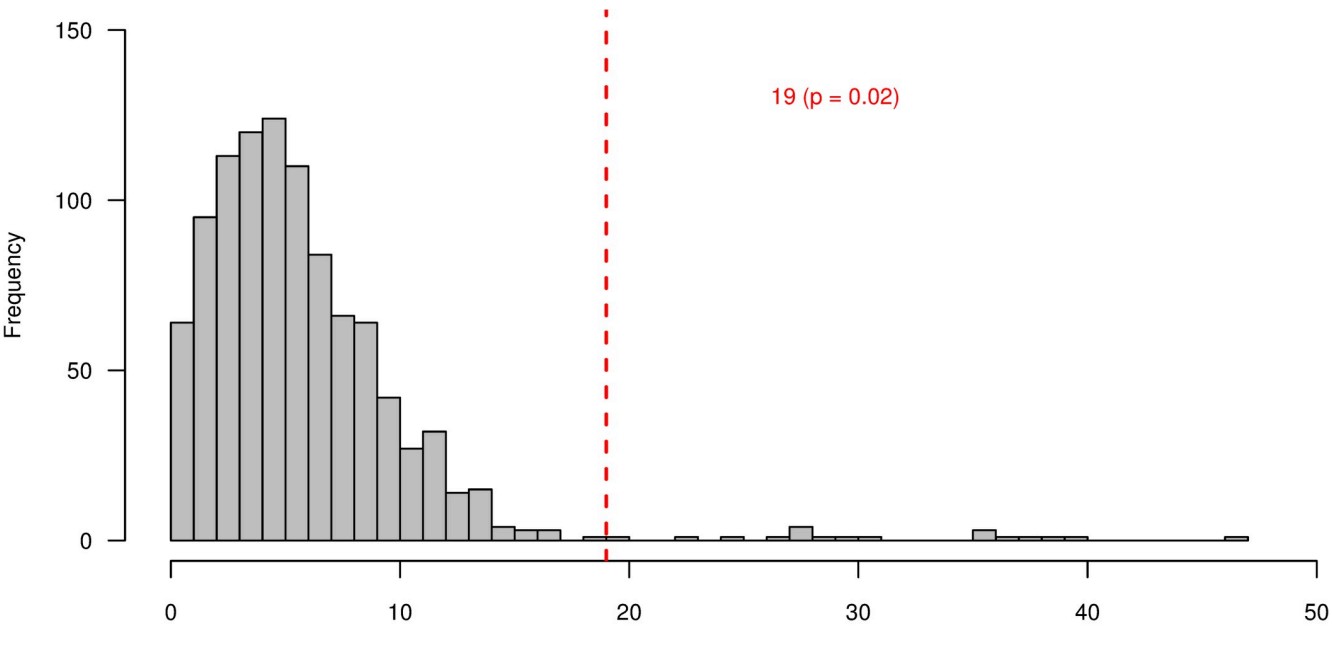

**Fig 6. Frequency distribution of the number of SNPs with near-independent significant effect on stature (P < 5x10-8) in 1,000 control tests for SMR-MT-C genes.** In each test, 121 cattle genes were sampled from a total of 13,865 genes that have human orthologs. Stratified random sampling was carried out, with genes stratified based on gene-size. The number of near-independent significant SNPs within and 100k either side of the sampled genes were identified. The vertical dashed line represents the number of near-independent significant SNPs (P < 5x10-8) associated with stature, identified within and 100kbp either side of the 121 cattle (SMR-MT-C) candidate genes, whose human orthologues show evidence of differential expression in 3 or more tissues. The p value represent the proportion of samples (out of 1,000) that have equal or higher level of enrichment for GWAS hits than the SMR-MT-C genes.

of orthologous genes in different species, although these studies focused on genes causing mendelian disorders, see [23] for a review. For a highly polygenic trait such as height, it is not yet established if the many genes underlying the trait in different species are conserved in their function, and if height-associated genes in one species can be used as prior information to identify the genes affecting height in a different species. In this study, we sought to establish if human height-associated orthologous genes, with mutations affecting height identified through GWAS, can be used as prior information to identify the genes and gene-associated variants that affect stature in cattle.

Due to LD, GWAS does not always result in the identification of the causal variants and/or genes for a given trait. In many cases, variants identified as significantly associated with a trait are proxies of unknown causal variants. One way of validating candidate genes identified from GWAS is to test if the genes are also associated with the same trait in a different species, given that LD pattern and thus the impact of LD on GWAS is different in different species. In our study, we identified a significant proportion (~39%) of cattle stature-associated orthologous genes that are also found to be associated with human height. Given that they appear as candidate genes for height in two species that separated over ~90 million years ago [24], the 30 overlapping genes cannot be considered an artifact of cryptic population structure. We found that 6 out of the 30 genes: ADAM12, SIK3, AXL, INSR, SPRED1 and PAPPA2 are involved in the Signaling by Receptor Tyrosine Kinases pathway, a pathway that is responsible for signaling of the growth hormone (GH) [25]. Two of the overlapping genes ADAMTS17 and ADAMTSL4 participate in the pathway: Defective B3GALTL causes Peters-plus syndrome (PpS). The Peters-plus syndrome is a recessive disorder that results in, among other things, short stature

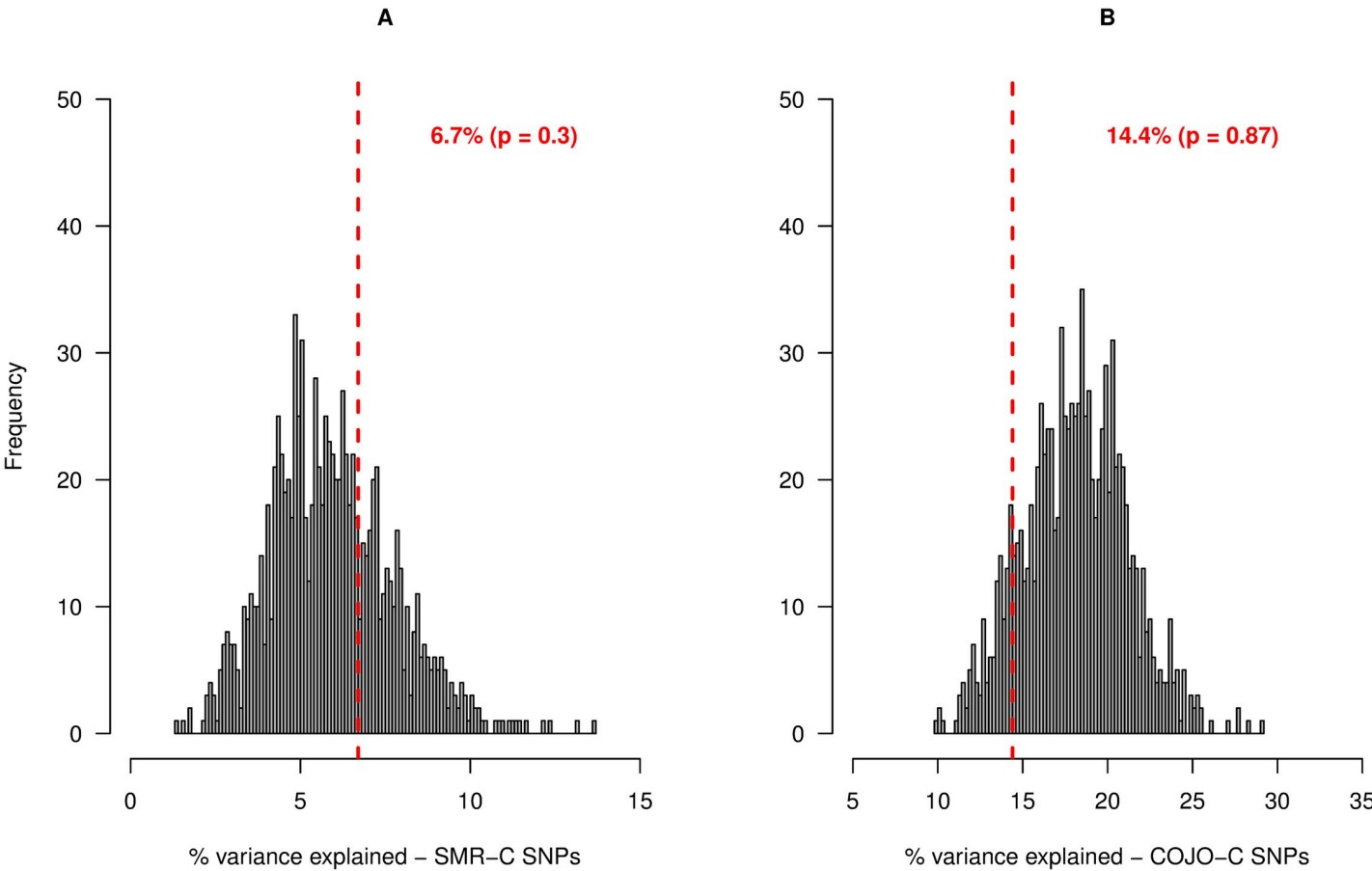

**Fig 7. Frequency distribution of the percentage of genetic variance in New Zealand Holstein population, explained by SNPs in 1,000 control tests for SMR-C SNPs.** In each test, the most significant SNPs for stature within a sampled gene-set, that had at least 20 copies of the minor allele in the population were used to construct the first GRM fitted in a GREML model. A second GRM was made with randomly sampled 50k SNPs (excluding SNPs in the first GRM) across the genome, to account for population structure. Figure A shows the result of control analysis for the SMR-C SNPs, in which 80 top SNPs in each gene set were used to construct the first GRM, while figure B shows the result of control analysis for the COJO-C SNPs, in which 391 top SNPs within each gene set was used to construct the first GRM. The red vertical lines represents the percentage of genetic variance explained by the candidate SMR-C and COJO-C SNPs, respectively.

and delay in growth [26]. Altogether, evidence suggests that height in humans and stature in cattle are biologically similar traits and that the orthologous genes associated with height in humans can be considered validated if they are also found to be associated with stature in cattle.

With respect to the identification of genes and gene-associated variants that affect stature in cattle, our results suggest that across-species prior information (results of GWAS for human height) is less informative than within-species prior information (results of GWAS for cattle stature). There are two reasons likely underpinning this finding. Firstly, the cattle (within-species) GWAS [12] was for a trait that is highly heritable. It is easier for GWAS to identify the signals underlying a highly heritable trait such as stature than for a lowly heritable trait e.g. fertility, given that at low heritability, there is larger environmental effect in the phenotype, which acts as noise around the genetic effects. In that sense, the benefit of using prior information from humans to identify genes and gene-associated variants for complex traits in cattle is expected to be higher for lowly heritable cattle traits than for highly heritable traits. For example, Costilla *et al.* [27] estimated the proportion of additive genetic variance for age at puberty in cattle explained by SNPs in and around cattle orthologous genes that were associated with

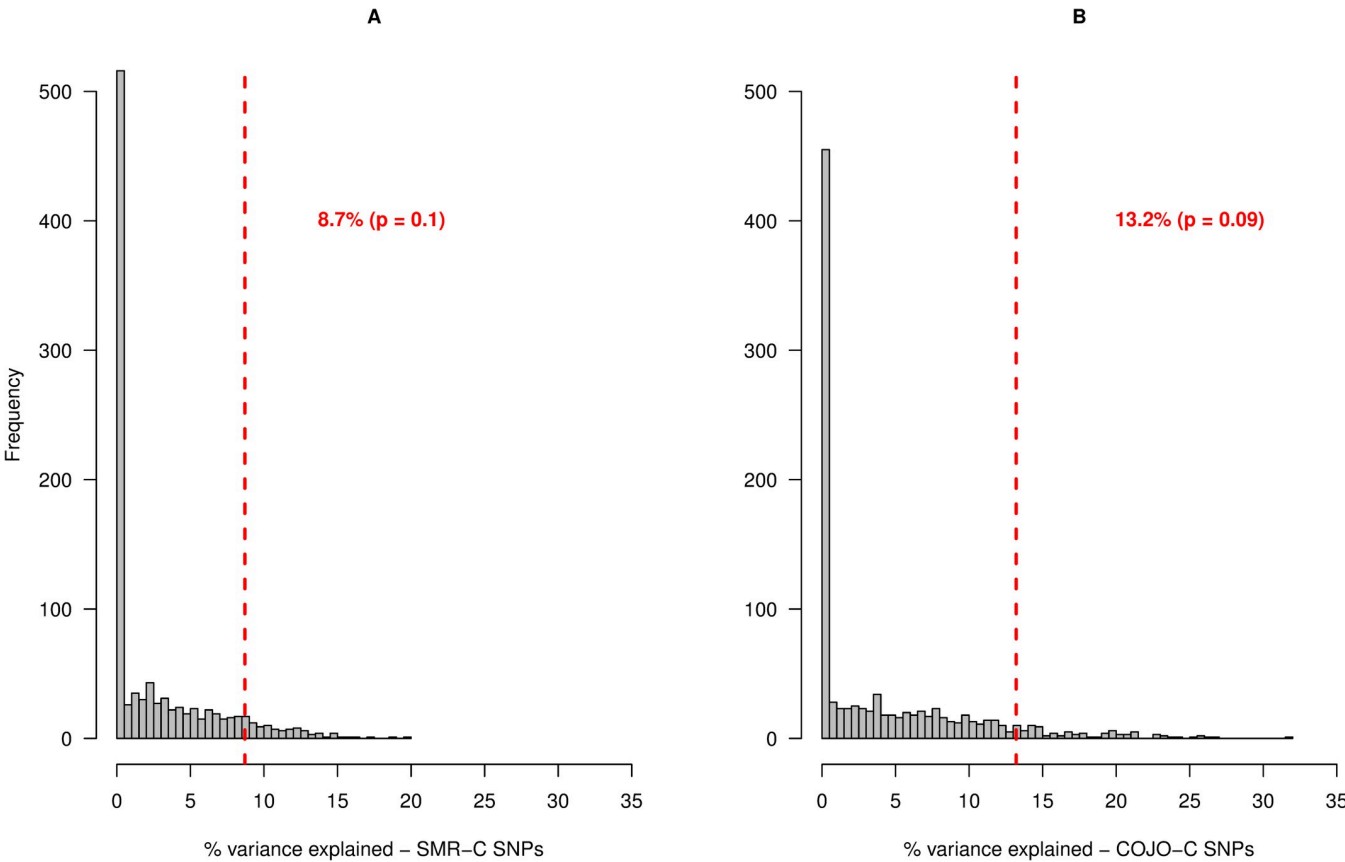

**Fig 8. Frequency distribution of the percentage of genetic variance in New Zealand Holstein population, explained by SNPs in 1,000 control tests for COJO-C SNPs.** In each test, SNPs were sampled randomly (matched by allele frequency to candidate SNPs) within a sampled gene-set (5 SNPs were sampled per gene, within each gene-set). SNPs had at least 20 copies of the minor allele in the validation population. Each sampled SNP set was used to construct the first GRM. A second GRM was made with randomly sampled 50k SNPs (excluding all SNPs in the first GRM) across the genome, to account for population structure. Figure A shows the result of control analysis for the SMR-C SNPs, in which 1,777 random SNPs in each gene set were used to construct the first GRM, while figure B shows the result of control analysis for the COJO-C SNPs, in which 5,781 random SNPs within each gene set was used to construct the first GRM. The red vertical lines represents the percentage of genetic variance explained by the candidate SMR-C and COJO-C SNPs, respectively.

age at menarche in humans. They showed that the SNPs in and around the candidate genes explained 11.2% of the total additive genetic variance for age at puberty in cattle, which was about twice the variance explained by SNPs in random gene-sets of similar size and SNP number.

Secondly, the within-species GWAS was large in terms of sample size and therefore was powerful enough such that the human prior information does not provide additional information. However, the GWAS study of [12] is an exceptional collaboration and data sharing in the livestock genetics community. An important factor for the success of the collaboration was that stature is generally not considered an economic trait, and in that sense, data on stature is less sensitive to share. Studies as large as that of [12] for economically important traits, especially those that are difficult or expensive to measure, are less common. For these kinds of traits, GWAS will most likely be underpowered, given that they are carried out at population level, where, in most cases, sample sizes are limiting. Identification of the genes and gene-associated variants that affects such traits will be aided by the integration of the huge resources, namely, GWAS results that are publicly available for complex traits in humans. Furthermore, for traits in cattle where powerful GWAS are not possible for whatever reason, using prior

information from human GWAS from similar traits will likely aid the identification of the underlying genes and the associated variants.

The method of determining the genes that underlie height in humans is very important, as this may determine whether or not the orthologs affects height in other species. In this study, we focused on two sets of orthologous genes that were determined to have an effect on human height. The COJO gene set are those that were prioritized as encompassing or close to the top GWAS signals for height in humans from [10]. The cattle orthologs of these genes were more enriched for cattle stature signals and captured more genetic variance for stature than random sample of genes in the cattle genome. The second set of genes were those that were prioritized based on the summary-based mendelian randomization analysis (SMR) in [10]. These are genes whose expression levels and association with height are assumed to be controlled by the same causal variant, and in that sense, assumed to be the most functionally relevant genes for human height. In our analysis, however, the cattle orthologs of the SMR-based human genes were not more enriched for cattle stature GWAS signals than random sets of genes from the cattle genome. There are two possible explanations for this observation. First, some genes are known to show tissue specific gene expression [28]. For their SMR analysis, Yengo *et al.* [10] used the GTEx-v7 database [29] that contains expression data of several genes in multiple tissues, not necessarily height related tissues. It is possible, therefore, that some of the genes showed differential level of gene-expression only in specific tissues that are not related to human height. The orthologs of such genes are less likely to affect height in other species. In contrast, genes with consistent differential expression in multiple tissue are more likely to maintain their effect across species. Our result, although only marginally significant (p = 0.02), suggests that to be the case. Given that the specific tissues that are relevant for specific traits of interest are not always known, the results of SMR analysis that uses gene expression data in only a single tissue, should be taken with caution. The second possible explanation might just be that the cattle meta-GWAS analysis of Bouwman *et al.* [12] was not powerful enough to detect important cattle stature variants within the cattle orthologs of the SMR-based human genes.

Potentially, GWAS for complex traits in humans could be utilized to aid precision genomic prediction in livestock. For example, knowledge about the position of candidate genes from GWAS in humans or the proportion of genetic variance for the trait that is explained by variants in or around the candidate genes, can be used to specify priors for genomic prediction models in livestock. A non-linear genomic prediction model that can be used for this purpose is the BayesR by segment (BayesRS) model [30]. In BayesRS, variant effects are assumed to belong to a mixture of four normal distributions, with the different distribution reflecting the different proportions of genetic variance explained by different genomic segments. Effectively, the BayesRS model can be specified such that the prior probability for a given variant depends on whether it is within or nearby an important gene identified from a human GWAS. An extended BayesRS model is the BayesRC model [31], which also allows for the explicit incorporation of prior knowledge in GP. In BayesRC, information on genes or variants from a human based GWAS can be used to assign variants into two or more classes (C). Other methods of using information from GWAS in humans for genomic prediction in livestock include weighting variants differently based on information from human GWAS in the construction of GRMs e.g. [32, 33], and fitting multiple GRMs in BLUP, with the different GRMs made from different classes of markers e.g. [34–36].

Altogether, our results suggest that human height and cattle stature are biologically similar traits. For the purpose of identifying the genes and gene-associated variants that affect stature in cattle, prior information from current human-height GWAS, does not provide more benefit than what a powerful GWAS for cattle-stature already provides. We must state that our

findings cannot easily be generalized to other complex traits. This is especially so for traits that have been under strong artificial selection in cattle, given that such traits might have evolved differently and rapidly than in humans. More research is needed to make such a generalization.

## Materials and methods

### Data

The summary level results from meta-GWAS for human height and cattle stature and other data sets used in the study are summarised in Table 1.

### Comparison of gene characteristics in humans and cattle

Many genes are conserved across species, therefore, comparison of traits across humans and cattle can be made at the gene level. For this study, we utilized information on orthologous genes between humans and cattle. Out of the 24,617 cattle genes identified in the UMD3.1 cattle genome build [37] and using Biomart Ensemb [38], we identified 13,742 genes that have orthologs in humans and passed quality control (QC). The QC thresholds used were: identity of the cattle gene to human gene > 70%; cattle gene order conservation score > 70%; cattle gene orthology confidence = 1, orthology type: 1 to 1.

Within each species (humans and cattle), we estimated the correlation between gene size and LD score of orthologous genes to see if longer genes have higher or lower level of LD than shorter genes. Furthermore, we estimated the correlation between human gene-size and cattle gene-size for the orthologous genes, and did the same for gene-based LD score. We calculated gene-size by subtracting the start, in base pairs (bp), from the end position of the gene and divided the value by 1,000 to obtain gene size in kilo base pairs (kbp). The start and end positions of a gene were defined as the beginning of the 5' UTR and the end of the 3' UTR of the gene, respectively. For humans, we used already calculated LD scores on UK10K [39] whole genome sequence variants. The LD score for each SNP was calculated as the sum of LD ($r^2$) between the SNP and all other SNPs in a 200kbp window, with 100kp overlap between two adjacent segments. The mean LD score of a gene was calculated as the mean LD score of all the SNPs within the gene. For cattle, we used GCTA [40] to calculated the LD scores for sequence variants on individuals from the Run4 of the 1,000 bulls genomes project dataset [41] that have 3 or more copies of the minor allele. The LD score for cattle SNPs was calculated with the same specifications as was used for human LD score calculations.

A high correlation between the effect of orthologous genes on human height and cattle stature may be an indication that the genes perform similar function across the species. Using the

**Table 1. A summary of all data sets used in the study.**

| Species | Data type | Sample size | # of SNPs | Reference |
|---------|-----------|-------------|-----------|-----------|
| Human | GWAS summary-level result | 253,288 | 2,550,858 | [9] |
| Human | GWAS summary-level result | 693,529* | 2,334,001 | [10] |
| Cattle | GWAS summary-level result | 58,265 | 25.4 million | [12] |
| Human | Individual-level imputed genotypes | 8,552 | 16,652,994 | [43] |
| Cattle | Individual-level whole genome sequence genotypes | 1,147 | 30.3 million | [41] |
| Human | LD scores | 10,000 | 17,613,557 | [39] |
| Cattle | Individual-level imputed genotypes and phenotype for stature | 975 | 14,341,737 | [34] |

*Mean sample size.

fastBat function in GCTA [42], and GWAS summary level results, we estimated the aggregated effect of all the SNPs within each orthologous gene on the phenotype (human height and cattle stature), which we refer to as gene-effect. We then calculated the correlation between the effect of the genes in both species. For the fastBat analysis to estimate the effect of the orthologous genes on human height, we used summary-level GWAS result from [10] and individual level imputed genotype data on individuals from the Health and Retirement Study [43] as LD reference. Similarly, for the fastBat analysis to estimate the effect of the orthologous genes on cattle stature, we used the summary-level result from [12] and the full sequence data on all individuals from Run4 of the 1,000 Bull Genomes Project [41] as LD reference. For human data, it was shown previously that the fastBat analysis loses power when SNPs are in very high or perfect LD, hence the recommendation to apply LD threshold of 0.9 [42]. We therefore used an LD threshold of 0.9 in the fastBat analysis of human data and set a lower LD threshold of 0.5 for the fastBat analysis of cattle data, given the high average LD between SNPs in the cattle genome. Outputs from the fastBat analysis were gene-based chi-squared (calculated as the sum of chi-squared values for all the SNPs within and 50kbp either side of a gene), and the corresponding gene-based p-value that indicate whether or not the effect of a gene on height (or stature in cattle) is significantly different from zero.

## Overlap of height-associated genes in humans and cattle

We estimated the number of orthologous genes in or close to GWAS peaks for human height that are also located in or close to GWAS peaks for stature in cattle, and tested if the overlap is more than can be expected by chance. Using the Biomart Ensembl tool, we identified the genes potentially affecting stature in cattle as those whose start or end position in the genome are located within 10kbp of the physical positions of the 163 lead SNPs for stature as reported in [12]. We identified 77 orthologous genes in total. Similarly, we identified a total of 368 human orthologous genes whose start or end positions in the genome are within 10kbp of the physical positions of the 649 lead SNPs for height as reported in [9]. We then checked the number of the 77 potential cattle genes that are part of the 368 potential human height genes.

Given the larger sample size used by [10], we wanted to know if there is a stronger overlap between the 77 cattle genes from [12], and human height genes from [10], as against those from [9]. To do this, we used the Biomart Ensembl tool to identify all human orthologous genes whose start or end positions in the genome are within 10kbp of the physical positions of the 3,290 near independent significant SNPs from [10], 1,553 genes in total. We then checked the number of the 77 potential cattle stature genes that are part of the 1,553 potential human height genes and tested if the number is more than can be expected by chance. In both cases, we used Fisher's exact test.

There is a chance that genes are enriched for GWAS signals and are identified as important genes for height in both humans and cattle, simply because they are large genes [44].Thus, we implemented the Cochran-Mantel-Haenszel test [45, 46] to test for the association between cattle stature genes and human height genes, while at the same time, accounting for a possible confounding effect of gene size. We implemented the test only for the overlap between the 77 cattle stature genes identified from [12], and human height genes identified from [10].

## Enrichment for cattle stature GWAS signals in and around human height-associated genes

The aim in this section was to quantify the usefulness of information from human height GWAS in the prioritization of stature genes in cattle, an analysis similar to [18]. Specifically, we tested the hypothesis that cattle genes prioritized based on information from a GWAS on

human height are more enriched for cattle stature GWAS signals than random sets of genes from the cattle genome. For the analysis, we used two different sets of human-height genes. First, we used the 610 human height genes identified by [10] using the Summary-based Mendelian Randomization (SMR) (Zhu *et al.* 2016) analysis [19]. The SMR analysis identifies genes whose expression level and association with the trait are assumed to be controlled by the same causal variant. We identified 370 cattle orthologs that will henceforth be referred to as SMR candidate (SMR-C) genes. The number of near-independent significant SNPs for cattle stature within and around the 370 SMR-C genes were then identified. To test if this number is more than can be observed with a random set of 370 cattle genes, we implemented a validation analysis, using randomized permutation test as follows: all 13,742 cattle genes that have human orthologs were stratified into 10 quantiles based on gene-size, number of SNPs within a gene or gene-based LD score, with the genes stratified separately for each criterion. In each quantile, we identified the number of SMR-C genes and subsequently sampled the same number of control genes from the quantile. Using this method, we sampled 370 control genes 1,000 times. We identified the number of near-independent SNPs within and 100kbp, 50kbp or 5kbp either side of the genes in each sampled control gene-set and checked if the number is more, equal or less than was observed within and 100kbp, 50kbp or 5kbp either side of the 370 SMR-C genes, respectively. For the GWAS signal enrichment analysis, near-independent significant SNPs for stature refers to SNPs obtained following a clumping analysis in plink 1.9 [47], using the summary level result from [12] and the fully sequenced individuals from Run4 of the 1000 Bull Genomes Project [41] as LD reference. The following parameters thresholds were set in the clumping analysis: significance level for index SNPs: 5x10-8; secondary significance level for clumped SNPs: 5x10-8; LD threshold for clumping: 0.2; physical distance threshold for clumping: 0.5mbp.

Secondly, we mapped all the 3,290 near-independent significant SNPs affecting human height identified through a conditional and joint effect (COJO) analysis by [10] to human genes, using Biomart Ensemb and the GRCh37 human reference genome. We identified 1,184 cattle orthologs of the COJO-based human height genes. These 1,184 cattle orthologous genes will henceforth be referred to as COJO candidate (COJO-C) genes. We carried out the same validation analysis as described for the SMR-C genes, with 1,184 control genes sampled 1,000 times.

In addition to possible differences in the usefulness of human prior information used to prioritize the SMR-C and the COJO-C cattle gene sets, respectively, difference in gene sample size (370 vs 1,184) can result in a difference in the level of enrichment for GWAS signals. To account for the difference in gene sample size, we took a sample of 370 genes from the 1,184 COJO-C genes 100 times, and checked the number of near-independent SNPs within and 100kbp either side of the genes in each sample. We then checked if the number of near independent SNPs for stature in each sample is more or less than was observed within and 100kbp either side of the 370 SMR-C genes.

Furthermore, we tested if the level of enrichment for GWAS hits within the SMR-C genes is stronger with more stringent selection criterium, namely, evidence of differential expression in multiple tissues. To do this, we identified 182 out of the 610 human height genes prioritized by [10] through SMR, that have differential gene expression evidence in 3 or more tissues. We located the cattle orthologs of the 182 genes, with a total of 121 remaining after QC. These 121 cattle orthologs genes will henceforth be referred to as: SMR evidence in multiple tissues candidate (SMR-MT-C) genes. We implemented a validation analysis as earlier described, to test if the number of near-independent SNPs within the 121 SMR-MT-C genes are more than can be observed within and around sampled control genes. In this case, we used stratified random sampling to sample 121 control genes 1,000 times, with genes stratified only based on gene

size, as this gives similar results compared to when genes are stratified based on gene-based LD score or SNP density.

## Accuracy of cattle variants selection using within- and across-species prior information

The aim in this section was to quantify the accuracy (in terms of the proportion of genetic variance explained for stature), of pre-selecting cattle SNPs based on GWAS results for height in humans. For the analysis, deregressed estimated breeding values (dEBVs) for stature and effective daughter contributions (EDCs), for 975 New Zealand Holstein (NZH) bulls was used. The EDCs were used as weights for the dEBVs in all analyses. To estimate the proportion of genetic variance explained by SNPs in the NZH population, a univariate GREML model with two genomic relationship matrices (GRMs) was fitted using the MTG2 software [48] follows:

$$\mathbf{y} = 1\mu + \mathbf{W}\mathbf{g_1} + \mathbf{W}\mathbf{g_2} + \mathbf{e},$$

where $\mathbf{y}$ is a vector containing the phenotypes (dEBVs of the NZH bulls), $\mu$ is the mean, $\mathbf{W}$ is an incidence matrix linking phenotypes to genetic effects, $\mathbf{g_1}$ and $\mathbf{g_2}$ are vectors of genetic effects corresponding to the two GRMs fitted simultaneously in the model, and $\mathbf{e}$ is a vector of residual effects. In this study, the first GRM ($\mathbf{GRM_1}$) was made using the SNPs of interest and the second GRM ($\mathbf{GRM_2}$) was made from 50k SNPs that were randomly sampled across the genome, excluding all SNPs in $\mathbf{GRM_1}$, and meant to account for possible population structure. Genetic effects in $\mathbf{g_1}$ and $\mathbf{g_2}$ are assumed to be normally distributed as $\mathbf{g_1} \sim N(0, \mathbf{GRM_1}\sigma_a^2)$ and $\mathbf{g_2} \sim N(0, \mathbf{GRM_2}\sigma_a^2)$, where $\sigma_a^2$ is the genetic variance for stature. Residual effects were assumed to be normally distributed as $\mathbf{e} \sim N(\mathrm{O}, \mathbf{D}\sigma_e^2)$, where $\sigma_e^2$ is the residual variance and $\mathbf{D}$ is a diagonal matrix that contains the inverse of EDCs, which were used as weights for the dEBVs. SNPs were included in the analysis that had at least 20 copies of the minor allele in NZH population. Thus, MAF threshold was set to $\frac{20}{975} * 2 = 0.04$. We tested two hypotheses as follows:

**Hypothesis 1**: Cattle SNPs that are pre-selected based on a combination of two information sources: 1) genes identified from GWAS results for human height, and 2) Cattle stature GWAS significance level, will explain more genetic variance for stature than SNPs that are pre-selected based from random cattle genes based on their significance level in a GWAS for cattle stature. To test this hypothesis, we identified the near-independent significant SNPs for stature in and 100kbp either side of the SMR-C and COJO-C genes, that segregate in NZH (80 SMR-C SNPs, 391 COJO-C SNPs). We used all the identified candidate SNPs to make $\mathbf{GRM_1}$ in GCTA [40]. For the control analysis, we sampled 1,000 control gene sets, and for each control gene set we ranked the SNPs based on significance level for association with stature and sampled the top 80 (SMR) or 391 (COJO) SNPs. So each set of control SNP set had the same number of SNPs as the candidate SNP set and was used to calculate $\mathbf{GRM_1}$. The proportion of total genetic variance explained by each set of control SNPs were estimated and compared to the genetic variance explained by the candidate SNPs.

**Hypothesis 2**: Cattle SNPs that are pre-selected based on GWAS result for human height will explain more genetic variance for stature, than random set of SNPs from random cattle genes. In this case, the candidate SNPs were all the SNPs identified within and 100kbp either side of the SMR-C and COJO-C genes, respectively. We randomly sampled 5 SNPs per gene to avoid a large number of SNP in high LD with each other, given the long range LD in commercial cattle populations. For the SMR-C genes, this resulted in 1,777 unique SNPs that were segregating in NZH. For the COJO-C genes, the number of unique SNPs extracted was 5,781. Each of these SNP set was used to calculate $\mathbf{GRM_1}$. For the control analysis, the same number

of SNPs as the candidate SNPs (1,777 and 5,781 SNPs respectively) were randomly sampled from within and 100kbp around the genes in each of the 1,000 control gene-sets and the SNPs were matched for allele frequency to the candidate SNPs.

## Supporting information

**S1 Fig. Distribution of average marker density in the 10kbp windows of association with lead SNPs from Bouwman *et al*. [12], Wood *et al*. [9] and Yengo *et al*. [10] and the average marker density of genes in both cattle and humans.** Average marker density were computed as total number of markers in a window or gene divided by the length (in kbp) of the window or gene.
(TIF)

**S1 Table. Contingency table showing the overlap of cattle stature genes from Bouwman *et al*. [12] with human height genes from Wood *et al*. [9].** *Cattle genes with 1 to 1 orthologs in humans (after QC); **Cattle genes within 10kb of the 164 lead SNPs from Bouwman *et al*. [12] that also have orthologs in humans; ***Human height genes prioritized as those that overlap with, or are within 10kkp either side of the 649 lead SNPs for height in Wood *et al*. [9], that also have orthologs in cattle. The proportion 10/77 is more than can be expected by chance: Fisher's exact test (odds ratio = 5.5, p-value = 3.7e-05).
(DOCX)

**S2 Table. Names and coordinates of 10 orthologous that are associated with both human height based on results from Wood *et al*. [9] and cattle stature based on results from Bouwman *et al*. [12].**
(DOCX)

**S3 Table. Contingency table showing the overlap of cattle stature genes from Bouwman *et al*. [12] with human height genes from Yengo *et al*. [10].** *Cattle genes with 1 to 1 orthologs in humans (after QC); **Cattle genes within 10kbp of the 164 lead SNPs from Bouwman *et al*. [12] that also have orthologs in humans; ***Human height genes prioritized as those that overlap with, or are within 10kbp either side of the 3,290 lead SNPs for height in Yengo *et al*. [10], that also have orthologs in cattle. The proportion 30/77 is more than can be expected by chance: Fisher's exact test (odds ratio = 5.1, p-value = 3.1e-10).
(DOCX)

**S4 Table. Names and coordinates of 30 orthologous that are associated with both human height based on results from Yengo *et al*. [10] and cattle stature based on results from Bouwman *et al*. [12].**
(DOCX)

**S5 Table. Contingency table showing the overlap of cattle stature genes from Bouwman *et al*. [12] with human height genes from Wood *et al*. [9], with the average marker density of the genes in both species.** + Total number of markers in the genes divided by gene length (in 10kpb). 10kbps windows were used to avoid having values of less than 1 in the contingency table. *Cattle genes with 1 to 1 orthologs in humans (after QC); **Cattle genes within 10kb of the 164 lead SNPs from Bouwman *et al*. [12] that also have orthologs in humans; ***Human height genes prioritized as those that overlap with, or are within 10kkp either side of the 649 lead SNPs for height in Wood *et al*. [9], that also have orthologs in cattle. Fisher's exact test of non-random association in gene marker density in cattle and humans (p-value = 0.87).
(DOCX)

**S6 Table. Contingency table showing the overlap of cattle stature genes from Bouwman** *et al.* **[12] with human height genes from Yengo** *et al.* **[10], while accounting for gene size.** Mantel-Haenszel chi-squared (corrected for gene length) = 35.6, p-value < 2.4e-9, common odds ratio = 4.1.

(DOCX)

## Author Contributions

**Conceptualization:** Biaty Raymond, Chris Schrooten, Aniek C. Bouwman, Ben J. Hayes, Roel F. Veerkamp, Peter M. Visscher.

**Data curation:** Biaty Raymond, Loic Yengo.

**Formal analysis:** Biaty Raymond, Loic Yengo, Roy Costilla.

**Investigation:** Biaty Raymond, Loic Yengo, Roy Costilla, Aniek C. Bouwman, Ben J. Hayes, Roel F. Veerkamp, Peter M. Visscher.

**Methodology:** Biaty Raymond, Loic Yengo, Roy Costilla, Aniek C. Bouwman, Ben J. Hayes, Roel F. Veerkamp, Peter M. Visscher.

**Resources:** Chris Schrooten, Roel F. Veerkamp, Peter M. Visscher.

**Supervision:** Roel F. Veerkamp, Peter M. Visscher.

**Writing – original draft:** Biaty Raymond.

**Writing – review & editing:** Biaty Raymond, Loic Yengo, Roy Costilla, Chris Schrooten, Aniek C. Bouwman, Ben J. Hayes, Roel F. Veerkamp, Peter M. Visscher.

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
