## [Decision Letter · Decision Letter 0]

10 Jun 2020

Dear Dr Raymond,

Thank you very much for submitting your Research Article entitled 'Using prior information from humans to prioritize genes and gene-associated variants for complex traits in livestock' to PLOS Genetics. Your manuscript was fully evaluated at the editorial level and by an independent peer reviewer. The reviewer appreciated the attention to an important topic but identified some aspects of the manuscript that should be improved.

We therefore ask you to modify the manuscript according to the review recommendations before we can consider your manuscript for acceptance. Your revisions should address the specific points made by each reviewer.

[LINK]

Yours sincerely,

Jennie Pryce

Guest Editor

PLOS Genetics

Gregory Barsh

Editor-in-Chief

PLOS Genetics

I am happy to endorse the opinion of the reviewer of this manuscript. The paper requires only very minor editing before it is ready for publication. It's a very interesting read.

Reviewer's Responses to Questions

**Comments to the Authors:**

Reviewer #1: Minor points:

line 136 –“…. no significant (p<0.05) ….” The ‘p<0.05’ is redundant as the exact p value has been reported nearby. Please remove the (p<0.05).

line 386-390 – Please be specific about the coordinates of start and end of the gene. Assuming this includes beginning of 5’UTR till the end of 3’ UTR. If so, please include in the methods description.

Figure2&3 Please make sure the y axis range is the same for both cattle and human graphs.

Table S1. Is the density of markers in datasets used the same with regards the 10K window of association with lead SNPs? There is a chance that OR of orthologue genes on chips with different marker density and genome gaps, affect the total likelihood of associations with said phenotype. Adding the marker density of the 10Kbp associated window and its average per gene in both species could clarify this issue. If there is a significant density difference (could be assessed the same way by Fisher’s exact 2x1 table) per same orthologue gene, please include that in the result section lines 149-165. This point also lends itself to the gene size-stratification analysis that has already been done using Cochran-Mantel- Haenszel in the following section.

**Have all data underlying the figures and results presented in the manuscript been provided?**

Reviewer #1: Yes

PLOS authors have the option to publish the peer review history of their article (what does this mean?). If published, this will include your full peer review and any attached files.

Reviewer #1: Yes: Mazdak Salavati DVM, PhD, AFHEA

---

## [Editor Report · Decision Letter 1]

21 Jul 2020

Dear Dr Raymond,

We are pleased to inform you that your manuscript entitled "Using prior information from humans to prioritize genes and gene-associated variants for complex traits in livestock" has been editorially accepted for publication in PLOS Genetics. Congratulations!

Yours sincerely,

Jennie Elizabeth Pryce

Guest Editor

PLOS Genetics

Gregory Barsh

Editor-in-Chief

PLOS Genetics

Comments from the reviewers (if applicable):

**Data Deposition**

http://datadryad.org/submit?journalID=pgenetics&manu=PGENETICS-D-20-00564R1

**Press Queries**

---

## [Editor Report · Acceptance letter]

4 Sep 2020

PGENETICS-D-20-00564R1 

Using prior information from humans to prioritize genes and gene-associated variants for complex traits in livestock 

Dear Dr Raymond, 

We are pleased to inform you that your manuscript entitled "Using prior information from humans to prioritize genes and gene-associated variants for complex traits in livestock" has been formally accepted for publication in PLOS Genetics! Your manuscript is now with our production department and you will be notified of the publication date in due course.

With kind regards,

Kaitlin Butler

PLOS Genetics

On behalf of:
